# Assessing Antioxidant Properties, Phenolic Compound Profiles, Organic Acids, and Sugars in Conventional Apple Cultivars (*Malus domestica*): A Chemometric Approach

**DOI:** 10.3390/foods13142291

**Published:** 2024-07-20

**Authors:** Biljana Cvetković, Aleksandra Bajić, Miona Belović, Lato Pezo, Danka Dragojlović, Olivera Šimurina, Marijana Djordjević, Karin Korntheuer, Christian Philipp, Reinhard Eder

**Affiliations:** 1Institute of Food Technology in Novi Sad, University of Novi Sad, Bulevar cara Lazara 1, 21000 Novi Sad, Serbia; aleksandra.bajic@fins.uns.ac.rs (A.B.); miona.belovic@fins.uns.ac.rs (M.B.); danka.dragojlovic@fins.uns.ac.rs (D.D.); olivera.simurina@fins.uns.ac.rs (O.Š.); marijana.djordjevic@fins.uns.ac.rs (M.D.); 2Institute of General and Physical Chemistry, University of Belgrade, Studentski trg 12/V, 11158 Belgrade, Serbia; latopezo@yahoo.co.uk; 3Federal College and Research Institute for Oenology and Pomology, Wiener Street 74, 3400 Klosterneuburg, Austria; karin.korntheuer@weinobst.at (K.K.); christian.philipp@weinobst.at (C.P.); reinhard.eder@weinobst.at (R.E.)

**Keywords:** apple, cultivar, antioxidants, polyphenols, chemometric

## Abstract

This study analyzed the phenolic compounds, organic acids, sugars, and antioxidant activity in different conventional apple cultivars (*Malus domestica*) from the Serbian market. Polyphenol profiles, sugars, and organic acid contents were analyzed by HPLC, and antioxidant activity was examined by DPPH and FRAP. Notable findings included variations in phenolic compound presence, with certain compounds detected only in specific cultivars. ‘Red Jonaprince’ exhibited the highest arbutin (0.86 mg/kg FW) and quercetin-3-rhamnoside content (22.90 mg/kg FW), while ‘Idared’ stood out for its gallic acid content (0.22 mg/kg FW) and ‘Granny Smith’ for its catechin levels (21.19 mg/kg FW). Additionally, malic acid dominated among organic acids, with ‘Granny Smith’ showing the highest content (6958.48 mg/kg FW). Fructose was the predominant sugar across all cultivars. Chemometric analysis revealed distinct groupings based on phenolic and organic acid profiles, with ‘Granny Smith’ and ‘Golden Delicious’ exhibiting unique characteristics. Artificial neural network modeling effectively predicted antioxidant activity based on the input parameters. Global sensitivity analysis highlighted the significant influence of certain phenolic compounds and organic acids on antioxidant activity.

## 1. Introduction

Apples (*Malus domestica*) in the rose family (*Rosaceae*) are the third most widely consumed fruit globally—approximately 96 million tons were consumed in 2022 worldwide [1]. In Europe, the major apple producers are Italy, France, Poland, and Germany [2]. The share of fruit production in the total value of agricultural production in the Republic of Serbia is 11% (Strategy of Agriculture and Rural Development of the Republic of Serbia 2014–2024). Apples are the second most cultivated fruit species in Serbia, with an annual production of 500,000 tons [3]. There are over 7000 apple cultivars globally. However, a limited number of well-adapted and closely related cultivars dominate global apple production. According to a previous study, apple production is concentrated on four primary cultivars: ‘Fuji’, ‘Gala’, ‘Golden Delicious’, and ‘Red Delicious’ [4]. RD (’Red Delicious’), GS (‘Granny Smith’), RJ (‘Red Jonaprice’), F (‘Fuji’), G (‘Gala’), GD (‘Golden Delicious’), and I (‘Idared’) are the most present varieties on the Serbian market.

The versatility of apples is highlighted, indicating that they are consumed in different forms, such as fresh fruit, processed foods, ciders, juices, concentrates, and purees [5].

Eating a variety of fruits and vegetables, including apples, as part of a balanced diet can contribute to overall health and well-being [6,7,8]. Apples represent an important source of bioactive compounds, such as pectin, organic acid, dietary fibers, vitamins, oligo-saccharides, and triterpenes acids, and phenolic compounds, such as flavonols, monomeric and oligomeric flavanols, anthocyanins, and *p*-hydroxycinnamic and *p*-hydroxybenzoic acids [9]. Polyphenols contained in apples, such as quercetin, catechins, and anthocyanins, act as antioxidants. They help neutralize harmful free radicals in the body, reducing oxidative stress and potentially lowering the risk of chronic diseases [2]. Some research suggests that polyphenols in apples may have neuroprotective effects, potentially reducing the risk of neurodegenerative diseases, such as Alzheimer’s [10]. A flavonoid and antioxidant, quercetin is abundant in apples, especially in the skin. It has anti-inflammatory properties and may contribute to heart health [11]. Chlorogenic acid may have anti-inflammatory and anti-diabetic properties [12,13]. Unique to apples, particularly concentrated in the skin, phloridzin has been studied for its potential benefits, including bone health [14,15]. Apple contains five major groups of polyphenolic compounds: hydroxycinnamic acids (mainly chlorogenic acid), flavan-3-ols ((+)-catechin, (−)-epicatechin, and anthocyanidins), flavonols (primarily various quercetin glycosides), dihydrochalcones (with phloridzin being the most abundant), and anthocyanins (primarily cyanidin-3-galactoside) [16]. Active phenolics, such as chlorogenic acids, phloretins, epicatechins, quercetins, and procyanidin B2, have been identified as major antioxidants in apples [17]. Recent research has revealed significant variation in the phenolic compound content across different apple cultivars [16,18]. As the authors claim, for a whole fruit, ‘Idared’ contains moderate levels of catechin (0.99 mg/g DW), chlorogenic acid (1.77 mg/g DW), epicatechin (0.68 mg/g DW), and rutin (0.45 mg/g DW), with relatively low phloridzin content (0.34 mg/g DW). ‘Golden Delicious’ has moderate amounts of catechin (0.99 mg/g DW), chlorogenic acid (1.23 mg/g DW), and epicatechin (0.65 mg/g DW), with a higher concentration of rutin (0.91 mg/g DW) and phloridzin (0.38 mg/g DW), while ‘Granny Smith’ stands out for having a higher level of catechin (1.47 mg/g DW) and rutin (1.30 mg/g DW), with moderate amounts of chlorogenic acid (0.67 mg/g DW), epicatechin (0.97 mg/g DW), and phloridzin (0.22 mg/g DW). Fuji variety contains moderate levels of catechin (0.88 mg/g DW), high levels of chlorogenic acid (1.35 mg/g DW), epicatechin (0.75 mg/g DW), and rutin (0.96 mg/g DW), with a relatively low concentration of phloridzin (0.34 mg/g DW). Red Delicious features moderate amounts of catechin (1.05 mg/g DW) and epicatechin (1.06 mg/g DW), with lower levels of chlorogenic acid (0.71 mg/g DW), rutin (0.48 mg/g DW), and phloridzin (0.43 mg/g DW) [19]. The difference in phenolic compounds among apple varieties is due to their biosynthesis pathways. The enzyme dihydroflavonol reductase (DFR) produces anthocyanins, which give red apples such as Red Delicious and Idared their color. Yellow and green apples, such as Golden Delicious and Granny Smith, have different enzyme activities, leading to different phenolic profiles. Environmental factors, such as sunlight, also affect phenolic content [20]. The Republic of Serbia benefits from favorable natural conditions and extensive areas suitable for cultivating numerous fruit species and high-quality varieties [21]. Considering that each geographical area has its own specificities, some studies have been conducted on the standard varieties of apples grown in Serbia [22,23]. However, there remains a notable gap in the data for certain varieties, such as Jonaprince and Fuji, particularly concerning their polyphenol profile, sugar, and organic acid content.

Common organic acids in apples include malic acid, citric acid, and tartaric acid. Organic acids, particularly malic acid, contribute to the tart flavor of apples. These acids may aid digestion by stimulating the production of digestive enzymes and promoting a healthy gut environment. Acidity in apples is a key factor influencing their taste and consumer preference. The acidity levels can vary significantly depending on the variety. Apples with lower acidity are generally perceived as sweeter and more palatable, which enhances their acceptance among consumers. For example, the Golden Delicious variety is frequently mentioned for its notably low acidity, with some reports indicating an acidity level of 0.17% [24]. In contrast, varieties such as Granny Smith are known for their higher acidity, contributing to their tart flavor. Understanding these variations is important for both consumers and producers. The sugar content in apples is primarily composed of fructose, glucose, and sucrose. Sweetness and tartness can vary significantly between cultivars. Some apples are naturally sweeter, while others may have a more balanced or tart flavor profile. ‘Gala’ and ‘Golden Delicious’ have the highest total soluble solids of 15.63 and 16.63, respectively, and consequently a higher proportion of sugar [25]. Ripeness at harvest also affects sugar levels, as apples continue to sweeten after being picked. The differences in polyphenols, antioxidant content, organic acids, and sugars among apple cultivars can be quite diverse, as these characteristics are influenced by genetic factors, growing conditions, and post-harvest handling [26]. The aim of this research is to reveal the true content of polyphenols, organic acids, and antioxidant activity of apples from the Serbian market that are available to the consumers within the food supply chain. The chemometric approach was used to group the apple cultivars on the basis of their content of phenolic compounds, organic acids, and sugars. The anticipation of antioxidant activities is directly correlated with the quantification of polyphenolic compounds in a given matrix [27]. Polyphenols, characterized by multiple phenol units, exhibit potent antioxidant properties by donating hydrogen atoms or electrons to neutralize reactive oxygen species [28]. Empirical studies consistently demonstrate a positive relationship between the polyphenol concentration and overall antioxidant capacity, as evaluated by assays, such as DPPH, ABTS, and FRAP [29]. This correlation is particularly significant in plant-based foods, where polyphenolic content serves as a reliable biomarker for potential health-promoting effects [30]. Similar correlations have been presented in apple cultivars’ antiradical capacity determinations [31,32]. Therefore, accurately measuring polyphenol levels is crucial for predicting the antioxidant efficacy of various dietary and pharmaceutical products [33]. To the best of our knowledge, such predictions have not been used for apple varieties so far.

An artificial neural network (ANN) model was used to predict the antioxidant activity of apples based on the content of individual phenolic compounds.

## 2. Materials and Methods

### 2.1. Plant Material

Fruits of seven apple varieties: RD (‘Red Delicious’), GS (‘Granny Smith’), RJ (‘Red Jonaprice’), F (‘Fuji’), G (‘Gala’), GD (‘Golden Delicious’), and I (‘Idared’), were purchased from Hiper Market Univer, Novi Sad Serbia. Apples were produced by local producers. The raw materials were sliced in quarters, directly frozen in liquid nitrogen, and freeze-dried for 72 h at 0.05 bar (Christ Alpha 1–4 LSC; Osterode am Harz, Germany). The homogeneous dry material was obtained by crushing the quarters of apples using a closed laboratory mill (IKA A.11, Staufen, Germany). The dried powders were kept in an ultra-freezer at −80 °C until extract preparation.

### 2.2. Extraction Procedure

For polyphenols and antioxidant activity analysis, the homogenized apple powder was leached three times with 70% CH_3_OH (2 × 3 mL and 1 × 2 mL), each time for 10 min in a cooled ultrasonic bath, centrifuged, and the supernatant was transferred to the 10 mL flask, filled up to 10 mL with distilled H_2_O, filtered at 0.2 μm, and filled into vials. For the organic acids and sugar analysis, the homogenized powder was leached three times with 10% C_2_H_5_OH (2 × 3 mL and 1 × 2 mL), each time for 30 min on the tube rotator, centrifuged, and the supernatant was transferred to the 10 mL flask, made up to 10 mL with distilled H_2_O, diluted at 1:25, and filtered through RP18 columns, then filled into vials and diluted again at 1:50 for sucrose, glucose, and fructose.

### 2.3. Identification and Quantification of Polyphenols

For the analysis of flavanols, flavonols, and phenolcarboxylic acids of the apple species, the slightly modified method of Vrhovsek et al. (1997) was used [34]. It was carried out on a Rapid-Resolution HPLC type 1200 (Agilent, Santa Clara, CA, USA) using an RP-C18 column (Poroshell 120 SB-C18 2.1 × 150 mm, Agilent, Santa Clara, CA, USA) coupled with DAD at 280 nm, 320 nm, and 362 nm. Mobile phase: eluent A: 0.5% formic acid, eluent B: methanol, gradient: 0 min 3% B, 0–13 min 3% B, 13–19 min 5% B, hold until 25 min, 25–34 min 6% B, 34–35 min 9% B, 35–52 min 10% B, 52–70 min 25% B, hold until 85 min, 85–100 min 40% B, and 100–105 min 90% B, with a 0.3 mL/min flow rate. Quantification was performed with external standard solutions by calculating the phenolic concentration from peak areas of the samples and corresponding external standards, and it was expressed in mg per kg fresh weight (mg/kg FW). Experiments were performed in triplicate.

### 2.4. Organic Acids and Sugar Analysis

The organic acids were analyzed using ion chromatography and chemical suppression with an ion conductivity detector, using a Dionex ICS 3000 series device (Dionex, Sunnyvale, CA, USA) according to the method described by Wendelin et al. (2018), following the company’s recommendations (Dionex AN 143) [35]. The anion exchange columns IonPac AG 11 and AS 11 from Dionex were used as the stationary phase. Eluent A: water, eluent B: 1.0 mM sodium hydroxide solution, eluent C: 100.0 mM sodium hydroxide solution, and eluent D: methanol. Gradient: 0–8 min 100% B, 8–16 min 67% A, 15% C, and 18% D, hold until 30 min, and 30–38 min 22% A, 60% C, and 18% D, with a 1.0 mL/min flow rate, according to the company’s recommendations (Dionex AN 143). The sugar spectrum (Dionex AN 122) was analyzed on a Carbo Pac10 (4 × 50/4 × 250) column with Dionex ICS 3000 series ion chromatography with electrochemical detection using the ED 40 Gold Electrode, Ag/AgCl Reference Electrode (Dionex, Sunnyvale, CA, USA), according to the method of Wendelin et al. (2018). Two exchange chromatography columns (CarboPac PA10, 4 × 250, and 4 × 50 (Dionex, Sunnyvale, CA, USA) on an IC device (ICS 3000 series; Dionex, Sunnyvale, CA, USA)) were used for separation. Mobile phase: Eluent A: water, and eluent B: 90 mM sodium hydroxide solution. Gradient (Dionex work instruction AN 122): 0–3 min 20% B, 3–8 min 100% B, and hold up to 22 min. Flow rate: 1.0 mL/min. Experiments were performed in triplicate and the results are presented in grams per kilogram fresh weight (g/kg FW).

### 2.5. Total Phenolic Content

Total phenolic content (TPC) was determined according to method of Singleton, Orthofer, and Lamuela-Raventos (1999), adapted for a plate reader (Multiskan Ascent, Thermo Electron Corporation, Waltham, MA, USA) [36]. Here, 125 μL of 0.1 M Folin–Ciocalteu reagent was added to 25 μL of extracts. After 10 min, 100 μL of 7.5% w/v sodium carbonate was added, and the reaction mixture was incubated for 2 h. Absorbance was read at 690 nm after the incubation period. In order to eliminate the interferences, correction was prepared by replacing the volume of reagents with the same volume of distilled water. A standard curve was prepared for gallic acid, and total phenolic content was expressed as mg of gallic acid equivalents (GAE) per kg of apple fresh weight. Experiments were performed in triplicate.

### 2.6. Determination of Antioxidant Activity

#### 2.6.1. DPPH Radical Scavenging Activity

Determination of free radical scavenging activity was based on the monitoring of 2,2-diphenyl-1-picrylhydrazyl (DPPH) radical transformation in the presence of antioxidants according to Espin, Soler-Rivas, and Wichers [37]. The reaction mixture in the wells consisted of 10 μL of sample, 60 μL of DPPH solution, and 180 μL of methanol. The control contained methanol instead of sample, and correction contained 10 μL of sample and 240 μL of methanol. After 60 min of incubation in the dark at room temperature, absorbance was measured using a plate reader at 492 nm. DPPH radical scavenging capacities of the extracts were calculated using the following equation:% DPPH radical scavenging capacity = 100 − ((Asample − Acorrection) 100)/Acontrol(1)

The IC50 value is the concentration that could scavenge 50% of the DPPH radicals. Each sample was tested at five different concentrations to obtain the IC50 value, and experiments were performed in triplicate.

#### 2.6.2. Ferric-Reducing Antioxidant Power (FRAP)

The FRAP test was performed according to the modified procedure of Benzie and Strain [38]. The FRAP reagent consisted of 300 mM of acetate buffer (pH = 3.6), 10 mM of 2,4,6-Tris(2-pyridyl)-s-triazine (TPTZ) in 40 mM of HCl, and 20 mM of FeCl_3_, in the ratio of 10:1:1 (v:v:v). Then, 10 μL of sample, 225 μL of FRAP reagent, and 22.5 μL of distilled water were added in a 96-well plate. Extract was replaced by the same volume of distilled water in the control, and correction contained distilled water instead of FRAP reagent. Absorbance was measured after 6 min at 630 nm. Ascorbic acid was used to construct the standard curve, and the results were expressed as mg of ascorbic acid equivalents (AAE) per kg of apple fresh weight. Each analysis was performed in triplicate.

### 2.7. Statistical Analysis

#### 2.7.1. Chemometric Analysis

PCA was employed to unveil the relationships among variables exhibiting similar interactions between variables. The data generated from PCA of the seven apple samples: RD (‘Red Delicious’), GS (‘Granny Smith’), RJ (‘Red Jonaprice’), F (‘Fuji’), G (‘Gala’), GD (‘Golden Delicious’), and I (‘Idared’), were visually represented via biplots, highlighting phenolic content and fruit acid contents. The analysis was performed using StatSoft Statistica 2012, ver. 12 software by StatSoft Inc., Tulsa, OK, USA.

#### 2.7.2. ANN Modeling

An artificial neural network (ANN) model was employed to predict the exploratory values for DPPH and FRAP, according to the phenolics content. Normalization of both input and output data was carried out to enhance the performance of the ANN. The selection of a multi-layer perceptron model (MLP) with three layers (input, hidden, and output) was based on its proven capability to approximate nonlinear functions, making it a suitable choice for this study. The Broyden–Fletcher–Goldfarb–Shanno (BFGS) algorithm, employed for ANN modeling, served as the training algorithm [39,40]. The coefficients for the hidden layer were represented by matrices *W*_1_ and *B*_1_, while the coefficients for the output layer were represented by matrices *W*_2_ and *B*_2_ [41]:(2)$$Y=f_{1}\left(W_{2}\cdot f_{2}\left(W_{1}\cdot X+B_{1}\right)+B_{2}\right)$$
where *f*_1_ and *f*_2_ are transfer functions in the hidden and output layers, respectively, and *X* is the matrix of input variables.

#### 2.7.3. Sensitivity Analysis

Yoon’s interpretation method was employed to assess the impact of phenolics content on TPC, DPPH, and FRAP. This assessment utilized weight coefficients derived from the constructed ANN model [42]:(3)$$RI_{ij}(\%)=\frac{\sum_{k=0}^{n}\left(w_{ik}\cdot w_{kj}\right)}{\sum_{i=0}^{m}\left|\sum_{k=0}^{n}\left(w_{ik}\cdot w_{kj}\right)\right|}\cdot 100\%$$
where: *w*—weight coefficient in the ANN model, *i*—input variable, *j*—output variable, *k*—hidden neuron, *n*—number of hidden neurons, and *m*—number of inputs.

## 3. Results and Discussion

The content of individual phenolic compounds determined in apple cultivars is presented in Table 1. It should be noted that certain phenolic compounds were detected only in some cultivars. Namely, arbutin was not detected in ‘Red Delicious’, ‘Fuji’, and ‘Granny Smith’ cultivars, while the highest amount was determined in the ‘Red Jonaprince’ cultivar. Regarding the phenolic acids, hydroxybenzoic acid was not detected in any cultivar, while neo-chlorogenic acid was detected only in the ‘Golden Delicious’ cultivar. Contrary to the presented results, hydroxybenzoic acid was previously detected in ‘Golden Delicious’ and ‘Idared’ cultivars [43]. The highest amount of *p*-coumaric acid was determined in Idared apples (3.14 mg/kg FW) but it was not detected in ‘Granny Smith’ apples. Chlorogenic acid was the most abundant phenolic acid, which is in accordance with previous studies [44,45]. Cultivar ‘Idared’ was the richest source of this phenolic acid (162.72 mg/kg FW). The detected concentrations of chlorogenic acid in the presented study, when recalculated to dry weight, were similar to concentrations published in the study by Liaudanskas et al. [44] but higher than those published in the study by Geană et al. [43]. Gallic acid was also detected in all apple cultivars, similar to the study by Geană et al. but in lower concentrations [43].

Catechin, epicatechin, and procyanidin B2 were flavan-3-ols detected in all samples, while procyanidin B1 was not detected in any apple sample, although previous studies indicated its presence in apples [44]. Generally, procyanidin B2 was the most abundant flavonoid in all apple cultivars. Regarding the flavonols, quercetin glycosides were detected in all apple samples, but there were no statistically significant differences among the cultivars in terms of quercetin-3-galactoside, quercetin-3-glucoside, and quercetin-3-rutinoside content. ‘Red Jonaprince’ had the highest content of quercetin-3-rhamnoside. Phloridzin, a dihydrochalcone typical for apple flesh [45], was determined in the highest amount in the ‘Red Delicious’ sample (23.92 mg/kg FW). Generally, the concentrations of phloridzin determined in the presented study correspond to those determined by Liaudanskas et al. after recalculation [44].

The total phenolic content and antioxidant activity of apple cultivars are presented in Table 2. The cultivar ‘Red Jonaprince’ had the highest amount of total phenolic compounds (972.74 mg GAE/kg FW), while the cultivar ‘Granny Smith’ had the lowest (579.39 mg GAE/kg FW). Generally, the obtained TPC values were in the range previously reported for various apple cultivars in the literature [46,47]. Differences between results found in the literature can be explained by the fact that apples were grown in different climatic conditions [18]. Antioxidant activity was spectrophotometrically tested using the free radical DPPH, which has been widely adopted for assessing antioxidant properties in different types of fruits [48]. In terms of the DPPH IC50 value, there were no significant differences between the apple cultivars, except ‘Granny Smith’ apples, which had a significantly higher value, indicating its lower radical scavenging potential when compared to other cultivars. Regarding the FRAP test, there were no significant differences between the cultivars, although ‘Red Jonaprince’ had the highest amount and ‘Gala’ had the lowest amount of ascorbic acid equivalents. A previous study by Kschonsek et al. showed that the cultivar ‘Golden Delicious’ had the lowest content of vitamin C in its flesh, and in the presented study, it was the cultivar with the second lowest amount of ascorbic acid equivalents [17].

The content of individual organic acids in apple cultivars is presented in Table 3. Malic acid was the dominant organic acid in all apple cultivars, as expected. Its content varied significantly, with ‘Granny Smith’ having the highest content (6958.48 mg/kg FW) and ‘Red Delicious’ having the lowest content (2744.00 mg/kg FW), which follows a previous study [49]. Generally, ‘Granny Smith’ is perceived by sensory panels as one of the apple cultivars with the most pronounced acidic taste [50]. On the other hand, ‘Red Delicious’ was characterized by the highest amount of quinic acid, while ‘Idared’, also rich in malic acid, had the lowest amount of quinic acid. Lactic acid, while rather abundant in ‘Red Delicious’ apples, was completely absent from ‘Red Jonaprince’ and ‘Golden Delicious’ cultivars. Conversely, galacturonic acid was detected only in these two cultivars. There were no significant differences among the apples in terms of succinic and oxalic acid content.

Generally, concentrations of oxalic, citric, malic, quinic, and succinic + shikimic acids determined in the presented study were in the range published previously by Mignard et al. [4] for various cultivars, including ‘Fuji’, ‘Gala’, ‘Golden Delicious’, and ‘Red Delicious’. Additionally, the concentration of malic acid determined in ‘Fuji’ apples corresponded to that determined by Zupan et al. (4.4 g/kg), but it was much lower than those reported by Aprea et al. (12.9–27.3 g/kg) [46,50].

The content of individual sugars in apple cultivars is presented in Table 4. Fructose was the most abundant sugar in all apple cultivars, as expected, followed by glucose and sucrose [4]. ‘Granny Smith’ had the lowest fructose content and the second lowest glucose content. Galactose was not detected in any apple sample. Sorbitol, which was previously shown to have a significant positive correlation with the perceived sweetness of apples (Aprea et al.), showed no significant difference in its content among the apples [50]. Myo-inositol content varied significantly among the apple cultivars, with ‘Golden Delicious’ having the highest and ‘Red Jonaprince’ having the lowest content.

The concentrations of glucose, fructose, sucrose, and sorbitol obtained for the cultivars ‘Fuji’, ‘Granny Smith’, and ‘Golden Delicious’ generally corresponded to those reported in the previous study [50], with some exceptions. Namely, ‘Fuji’ samples analyzed in this study had a higher sorbitol content (6.0–12.7 g/kg), while the ‘Granny Smith’ sample had higher sucrose (47.7 g/kg) and lower fructose contents (37.5 g/kg). However, in another study, ‘Fuji’ apples had a lower sorbitol content (4.6 g/kg), closer to the value determined in the presented study [46]. These differences were not surprising since a previous study performed on 155 accessions in Spain, including ‘Fuji’, ‘Gala’, ‘Golden Delicious’, and ‘Red Delicious’, showed significant variability between minimum and maximum concentrations of sugars and sorbitol in apples grown in different climate conditions and geographic regions [4].

### 3.1. Chemometric Analysis of the Results of Phenolic Content

The PCA of apple samples’ phenolic content (Figure 1) demonstrated that the first two principal components captured 65.4% of the total variability among the 13 parameters (A—arbutin; GA—gallic acid; nCA—neo-chlorogenic acid; C—catechin; CA—chlorogenic acid; pCA—*p*-coumaric acid; pCB2—procyanidin B2; E—epicatechin; Q3ga—Qu-3-galactoside; Q3gl—Qu-3-glucoside; P—phloridzin; Q3ru—Qu-3-rutinoside; Q3rh—Qu-3-rhamnoside). In this analysis, certain phenolics, such as procyanidin B2 (contributing 17.6% to the total variance), epicatechin (17.3%), and quercetin-3-glucoside (8.7%), exhibited a positive influence on the first principal component (PC1). Conversely, chlorogenic acid (13.2%) and *p*-coumaric acid content (14.3%) demonstrated a negative influence on the calculation of PC1. Meanwhile, phenolics such as arbutin (19.6% of total variance), quercetin-3-galactoside (19.2%), quercetin-3-rutinoside (24.2%), and quercetin-3-rhamnoside (26.1%) displayed a positive effect on the second principal component (PC2).

‘Red Jonaprince’ was separated from the other apple cultivars by its high content of arbutin and quercetin-3-rhamnoside. ‘Red Delicious’ was distinguished by the highest gallic acid and high epicatechin content, but the highest epicatechin content was determined in ‘Granny Smith’ apples. This cultivar was also characterized by the highest catechin and procyanidin B2 content. Although different by their color, ‘Idared’, ‘Golden Delicious’, and ‘Fuji’ were grouped closely in the same quadrant by their profiles of phenolic compounds.

### 3.2. Chemometric Analysis of the Results of Fruit Acid and Sugar Content

The PCA of fruit acid and sugar content in apple samples (Figure 2) revealed that the first three principal components encompassed 78.4% of the total variability across 15 parameters (QA—quinic acid; LA—lactic acid; ShA—shikimic acid; GA—galacturonic acid; SuA—succinic acid; MA—malic acid; FA—fumaric acid; OA—oxalic acid; P—phosphate; CA—citric acid; MI—myoinosit; S—sorbitol; Gal—galactose; Glu—glucose; Fru—fructose; Suc—Sucrose). Notably, the contents of: galacturonic acid (contributing 8.3% to total variance, based on correlations), malic acid (13.8%), phosphate (13.2%), citric acid (19.4%), and sucrose (15.6%) exhibited a negative influence on the first principal component (PC1), while glucose (16.2%) influenced positively to PC1 coordinate calculation.

Meanwhile, the contents of lactic acid (13.0% of total variance, based on correlations) and shikimic acid (22.7%) had a positive impact on the second principal component (PC2), whilst the contents of galacturonic acid (7.3%), sorbitol (16.8%), and fructose (18.7%) showed a negative effect on the calculation of the PC2 component. Furthermore, fumaric acid content (8.7%) positively influenced the calculation of the third principal component (PC3), while the contents of quinic acid (19.5%), succinic acid (21.9%), and myo-inositol (20.4%) showed a negative influence on the calculation of the PC3 coordinate.

Looking at the PC1 and PC3 plot, ‘Granny Smith’ was separated from other apples by the highest malic and shikimic acid content, along with the lowest content of fructose. ‘Golden Delicious’ was distinguished from other apples by the highest sucrose and the lowest glucose content. On the other side, red cultivars, ‘Gala’, ‘Red Jonaprince’, and ‘Red Delicious’, were differentiated by high glucose and low sucrose content. ‘Fuji’ cultivar was distinguished by its high fumaric acid content. Similar layouts of the analyzed apple samples are seen on the PC1 and PC3 plot (Figure 2b), with some differences. Namely, ‘Gala’ and ‘Red Jonaprince’ were positioned very closely due to their high glucose content, and ‘Golden Delicious’ and ‘Idared’ were closer than on the PC1 and PC2 plot, because of their high phosphate content.

According to the correlation analysis (Figure 3), statistically significant correlations were observed between the content of citric acid and the contents of malic acid and phosphate (r = 0.837; r = 0.872; statistically significant at the *p* < 0.05 level), and negative correlations between the content of glucose and the contents of malic acid, citric acid, and myo-inositol (r = −0.786; r = −0.853; r = −0.797; statistically significant at the *p* < 0.05 level). A positive correlation between sorbitol and succinic acid (r = 0.797; *p* < 0.05) and negative correlation between fructose and shikimic acid (r = −0.812; *p* < 0.05) were observed. A positive correlation between sucrose and citric acid (r = 0.883; *p* < 0.01) and negative correlation between sucrose and glucose (r = −0.758; *p* < 0.05) were also observed (Figure 3).

TPC was negatively correlated with DPPH (r = −0.462; *p* = 0.035) and positively correlated with the FRAP value (r = 0.543; *p* = 0.011), as expected (Table 5). Regarding the individual phenolic compounds, *p*-coumaric acid content was negatively correlated with DPPH (r = −0.853; *p* = 0.015), and procyanidin B2 was positively correlated with DPPH (r = 0.834; *p* = 0.020). This can be explained by the fact that *p*-coumaric acid was not detected in ‘Granny Smith’ apples, and the highest procyanidin B2 concentration was measured in this cultivar (Table 1), characterized by the lowest antiradical activity. However, a previous study showed that procyanidin B2 content in ‘Granny Smith’ apple peel extracts had a strong correlation with DPPH radical scavenging activity [51].

### 3.3. ANN Modeling of Antioxidant Activity

The structure and outcomes of the artificial neural network model depend heavily on the initial assumptions made for matrix parameters (biases and weights), which are crucial for fitting the model to experimental data. The model’s performance is also affected by the number of neurons in the hidden layer. To address this, 100,000 runs with randomized topologies were conducted to eliminate random correlations from initial assumptions and weight initialization. The model attained its highest r^2^ value with eleven hidden neurons.

The optimized neural network models effectively generalized the experimental data, accurately predicting output from input parameters. Using an ANN model with 11 neurons (network MLP 12-11-2) resulted in high r^2^ values (1.000 for training). Detailed matrix *W*_1_ and vector *B*_1_ (bias row) results are provided in Table 6, while Table 7 presents elements of matrix *W*_2_ and vector *B*_2_ (bias) for the hidden layer.

The ANN model was used to calculate the value of the antioxidant parameters of DPPH and FRAP based on the content of individual polyphenolic compounds. This calculation should be adopted as an estimation (regardless of the r^2^ value being 1).

Through the ANN model, the influence of individual polyphenols on the antioxidant parameters DPPH and FRAP could be examined, which is further discussed in the global sensitivity analysis. It is important to point out that the previous study used spectral reflectance, color, and only one biochemical property to predict the antioxidant activity determined by the DPPH and FRAP assays [52].

In the article by Peter et al. [53], a first-order polynomial model was employed to estimate the main effects of polyphenols, carotenoids, and chlorophylls on ABTS. The model demonstrated a statistically insignificant deviation from the experimental data, corroborating its suitability. The evaluated values of r^2^ (0.990), RMSE (20.457), and χ^2^ (470.814) confirmed the model’s statistical significance and alignment with experimental results. In a study by Đorđević et al. [54], an ANN model was used to predict antioxidant activity in Montenegrin Merlot wine based on the polyphenol concentration. The ANN models exhibited strong predictive capabilities for antioxidant parameters, with r² values ranging from 0.923 to 0.976. A similar study by Stajčić et al. [55] developed an artificial neural network model to predict antioxidant activity based on the content of bioactive compounds (phenolics and carotenoids) in oven-dried (at 50 and 65 °C) and freeze-dried pumpkin waste after one year of storage. These models showed excellent predictive accuracy, with an r^2^ value of 0.999 during the training cycle for the output variables. According to the investigation by Nićetin et al. [56], an ANN model was developed to predict antioxidant activity based on the phenol content in celery root after osmotic dehydration in molasses, achieving an overall r^2^ of 0.812 and a low sum of squares (0.0005). The most significant influences on FRAP and ABTS were observed for luteolin, chlorogenic acid, kaempferol, gallic acid, chrysin, and catechin contents.

### 3.4. Global Sensitivity Analysis—Yoon’s Interpretation Method

The impact of 12 input variables (such as phenolics content: A, GA, C, CA, pCA, pCB2, E, Q3ga, Q3gl, P, Q3ru, and Q3rh) on DPPH and FRAP was examined. For the DPPH value, a positive influence was observed with the presence of A, C, pCB2, and Q3ga, with relative importance of +11.51%, +13.44%, +7.27%, and +9.82%, respectively. Conversely, the content of CA (−10.49%) and P (−26.99%) had the strongest negative relative impact (Figure 4). Similarly, the FRAP value was primarily influenced by GA content (+23.43%), with the most negative impact observed for pCB2 (−16.02%) and P (−16.11%; Figure 4). A previous study showed that flavonols have the major contribution to antioxidant activity in apple peel, while flavanols have a dominant contribution to antioxidant activity in apple flesh, with procyanidin B2 being one of the major contributors [17]. Since the whole apple was analyzed, it was logical for apple flesh to have a dominant influence on antioxidant activity. However, the chemical structure of the compounds had a stronger effect on free radical quenching than the amounts of individual polyphenols [57], which can explain the negative effect of some phenolic compounds on the antioxidant activity.

## 4. Conclusions

Conventional apple cultivars (*Malus domestica*) from the Serbian market showed great variation in composition and amounts of phenolic compounds, both individual and total. Smaller variation was observed for organic acids and sugars, where malic acid was the dominant organic acid, and fructose was the dominant sugar in all apple samples. Chemometric analysis showed that the ‘Granny Smith’ cultivar was separated from others by its lower antioxidant activity, higher content of acids, and lower content of sugars. On the other hand, red cultivars, mainly ‘Red Jonaprince’, were characterized by higher antioxidant activity and higher glucose content. Although the ANN model successfully related antioxidant activity, expressed as DPPH radical scavenging and ferric-reducing antioxidant power (FRAP), to the individual phenolic compounds, the obtained results have limitations. The first one is that the chemical structure of individual polyphenols is also important for antioxidant activity, not only their amount in apples. The second one is that only seven of the most popular cultivars were analyzed. Phenolic compounds, such as gallic acid, catechin, quercetin-3-galactoside (Qu-3-galactoside), and quercetin-3-glucoside (Qu-3-glucoside), had the highest relative influence on DPPH and FRAP values, indicating significant antiradical activity. In contrast, procyanidin (pCB2) negatively impacted antiradical activity. This finding suggests that future research should focus on the structural characteristics of polyphenolic compounds, not just their quantity, to better understand their influence. Future studies should also include more yellow and green cultivars to have a better representation of various profiles of phenolic compounds.

## Figures and Tables

**Figure 1 foods-13-02291-f001:**
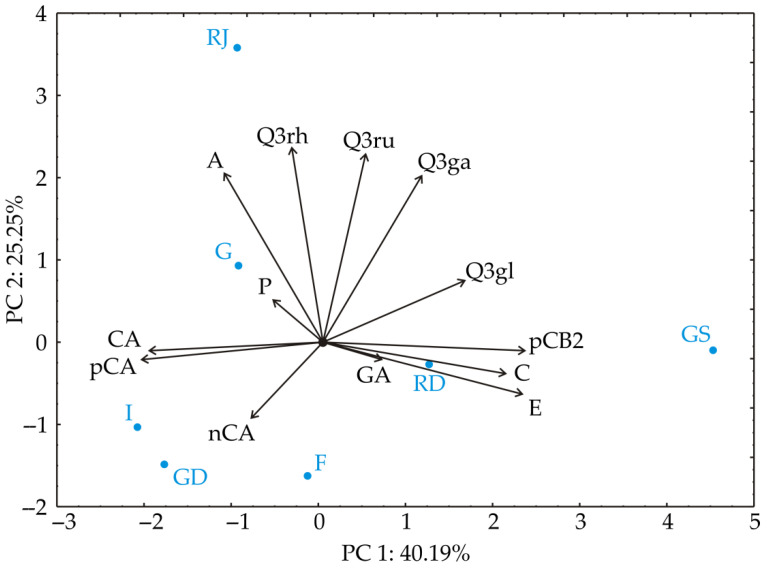
PCA ordination of phenolic content: projection in the PC1–PC2 plane. A—arbutin; GA—gallic acid; nCA—neo-chlorogenic acid; C—catechin; CA—chlorogenic acid; pCA—*p*-coumaric acid; pCB2—procyanidin B2; E—epicatechin; Q3ga—Qu-3-galactoside; Q3gl—Qu-3-glucoside; P—phloridzin; Q3ru—Qu-3-rutinoside; Q3rh—Qu-3-rhamnoside; G—Gala; RJ—Red Jonaprince; RD—Red Delicious; F—Fuji; GS—Granny Smith; I—Idared; GD—Golden Delicious.

**Figure 2 foods-13-02291-f002:**
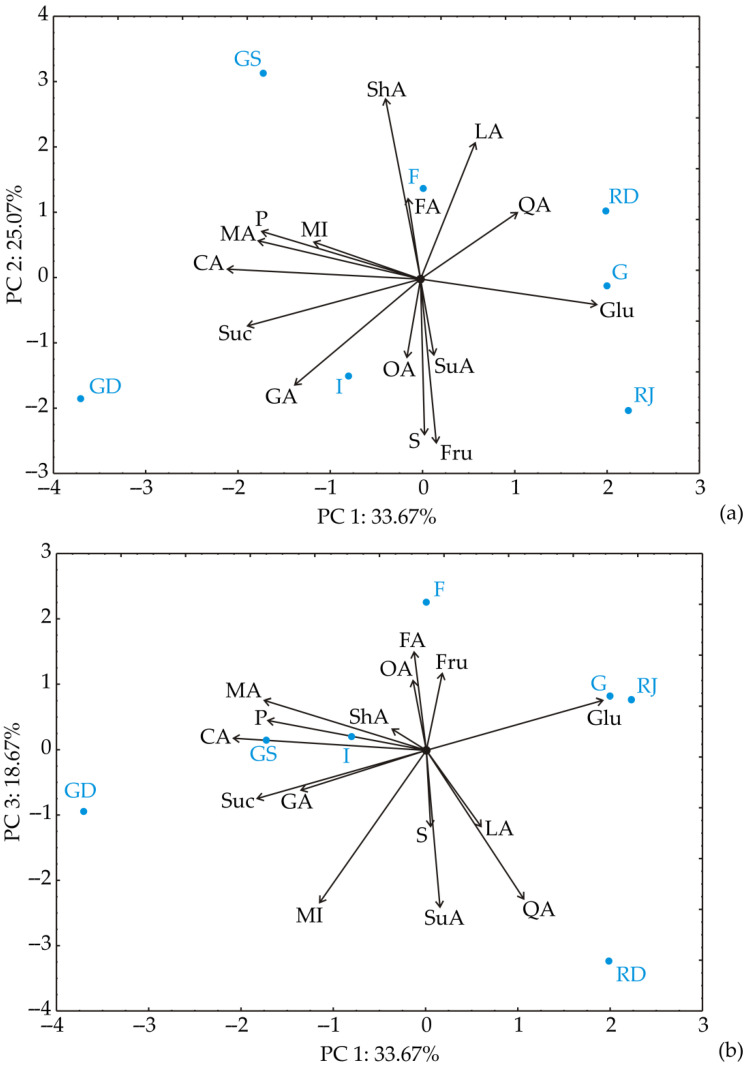
PCA coordination of fruit acids and sugars content: (**a**) projection in PC1-PC2 plane; (**b**) projection in PC1–PC3 plane. QA—quinic acid; LA—lactic acid; ShA—shikimic acid; GA—galacturonic acid; SuA—succinic acid; MA—malic acid; FA—fumaric acid; OA—oxalic acid; P—phosphate; CA—citric acid; MI—myo-inositol; S—sorbitol; Gal—galactose; Glu—glucose; Fru—fructose; Suc—Sucrose; G—Gala; RJ—Red Jonaprince; RD—Red Delicious; F—Fuji; GS—Granny Smith; I—Idared; GD—Golden Delicious.

**Figure 3 foods-13-02291-f003:**
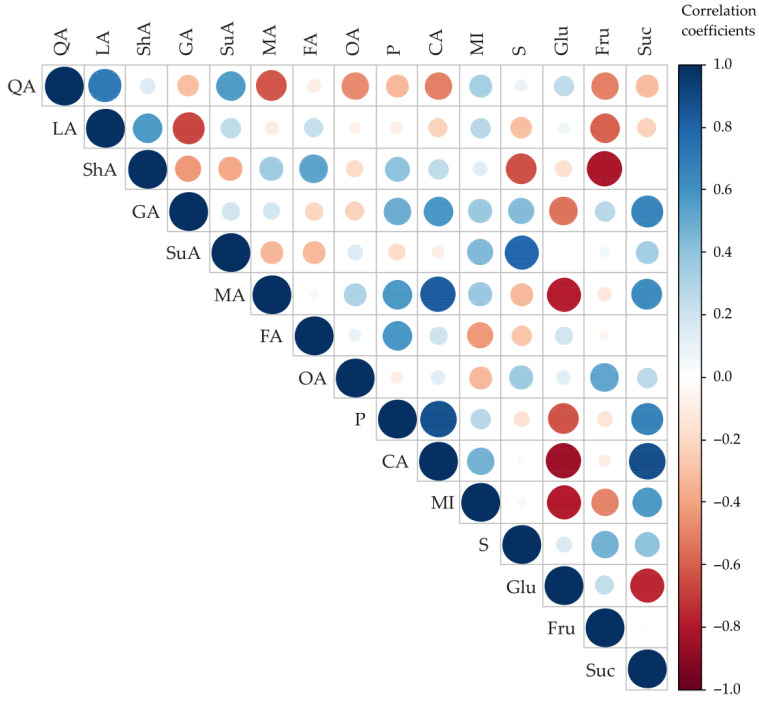
Correlation analysis of organic acid and sugar contents for conventional apple cultivars (*Malus domestica*) from the Serbian market. QA—quinic acid; LA—lactic acid; ShA—shikimic acid; GA—galacturonic acid; SuA—succinic acid; MA—malic acid; FA—fumaric acid; OA—oxalic acid; P—phosphate; CA—citric acid; MI—myo-inositol; S—sorbitol; Gal—galactose; Glu—glucose; Fru—fructose; Suc—sucrose.

**Figure 4 foods-13-02291-f004:**
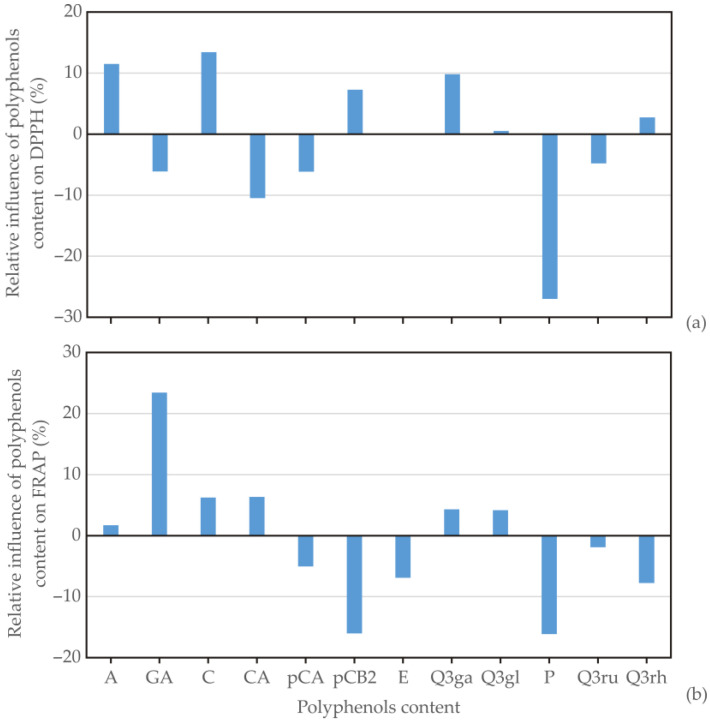
Relative influence of phenolic contents on (**a**) DPPH and (**b**) FRAP values for conventional apple cultivars (*Malus domestica*) from the Serbian market. A—arbutin; GA—gallic acid; nCA—neo-chlorogenic acid; C—catechin; CA—chlorogenic acid; pCA—*p*-coumaric acid; pCB2—procyanidin B2; E—epicatechin; Q3ga—Qu-3-galactoside; Q3gl—Qu-3-glucoside; P—phloridzin; Q3ru—Qu-3-rutinoside; Q3rh—Qu-3-rhamnoside.

**Table 1 foods-13-02291-t001:** Content of individual phenolic compounds of the conventional apple cultivars (*Malus domestica*) from the Serbian market.

Conc. mg/kg FW		Gala	Red Jonaprince	Red Delicious	Fuji	Granny Smith	Idared	Golden Delicious
**A**		0.24 ± 0.41 ^ab^	0.86 ± 0.58 ^b^	n.d.	n.d.	n.d.	0.21 ± 0.36 ^ab^	0.29 ± 0.49 ^ab^
Phenolic acids *	HBA	n.d.	n.d.	n.d.	n.d.	n.d.	n.d.	n.d.
	GA	0.17 ± 0.07 ^a^	0.14 ± 0.04 ^a^	0.56 ± 0.13 ^b^	0.16 ± 0.12 ^a^	0.18 ± 0.03 ^a^	0.22 ± 0.05 ^a^	0.10 ± 0.09 ^a^
	pCA	2.57 ± 0.53 ^cd^	1.72 ± 0.53 ^bc^	2.15 ± 0.42 ^c^	1.33 ± 0.07 ^b^	n.d.	3.14 ± 0.16 ^d^	2.36 ± 0.18 ^cd^
	CA	156.37 ± 46.01 ^cd^	94.93 ± 30.04 ^b^	75.90 ± 15.06 ^ab^	104.52 ± 5.22 ^bc^	30.24 ± 8.71 ^a^	162.72 ± 10.27 ^d^	97.49 ± 14.88 ^bc^
	nCA	n.d.	n.d.	n.d.	n.d.	n.d.	n.d.	0.17 ± 0.03
Flavan-3-ols	C	9.46 ± 4.10 ^bc^	3.02 ± 0.63 ^a^	13.02 ± 1.67 ^c^	5.57 ± 0.90 ^ab^	21.19 ± 1.48 ^d^	8.89 ± 1.64 ^bc^	4.02 ± 0.16 ^a^
	E	31.04 ± 4.79 ^a^	26.40 ± 4.13 ^a^	48.92 ± 6.21 ^b^	36.79 ± 4.46 ^a^	59.53 ± 7.65 ^b^	27.58 ± 2.35 ^a^	32.88 ± 2.82 ^a^
	pCB1	n.d.	n.d.	n.d.	n.d.	n.d.	n.d.	n.d.
	pCB2	64.17 ± 9.27 ^ab^	56.47 ± 9.91 ^a^	73.11 ± 5.93 ^ab^	58.38 ± 11.66 ^a^	92.96 ± 24.12 ^b^	54.92 ± 3.96 ^a^	58.92 ± 4.51 ^a^
Flavonol	Q3ga	6.96 ± 2.91 ^a^	9.14 ± 7.73 ^a^	8.25 ± 4.18 ^a^	3.17 ± 2.01 ^a^	8.06 ± 3.58 ^a^	4.68 ± 2.92 ^a^	3.87 ± 5.21 ^a^
	Q3gl	1.72 ± 0.50 ^a^	2.21 ± 1.48 ^a^	1.34 ± 0.49 ^a^	2.33 ± 2.78 ^a^	2.78 ± 0.92 ^a^	1.00 ± 0.10 ^a^	1.37 ± 1.05 ^a^
	Q3ru	7.52 ± 2.33 ^a^	7.93 ± 4.48 ^a^	5.90 ± 1.82 ^a^	3.13 ± 1.33 ^a^	5.88 ± 2.03 ^a^	4.27 ± 0.68 ^a^	2.85 ± 2.03 ^a^
	Q3rh	10.41 ± 5.91 ^ab^	22.90 ± 11.46 ^b^	5.69 ± 1.57 ^a^	4.40 ± 1.66 ^a^	8.19 ± 1.95 ^a^	4.96 ± 1.17 ^a^	8.79 ± 3.32 ^a^
Dihydrochalcone	P	18.97 ± 4.32 ^bc^	12.85 ± 2.42 ^ab^	23.92 ± 8.34 ^c^	8.25 ± 1.21 ^a^	5.58 ± 3.37 ^a^	13.00 ± 1.61 ^ab^	10.78 ± 2.67 ^ab^

* Data are expressed as means (*n* = 3) ± standard deviations. Lowercase letters within a row indicate statistically significant differences at *p* ≤ 0.05. Abbreviations: A—arbutin; HBA—hydroxybenzoic acid; GA—gallic acid; pCA—*p*-coumaric acid; CA—chlorogenic acid; nCA—neo-chlorogenic acid; C—catechin; E—epicatechin; pCB1—procyanidin B1; pCB2—procyanidin B2; Q3ga—Qu-3-galactoside; Q3gl—Qu-3-glucoside; Q3ru—Qu-3-rutinoside; Q3rh—Qu-3-rhamnoside; P—phloridzin; n.d.—not detected.

**Table 2 foods-13-02291-t002:** Total phenolic content and antioxidant activity of the conventional apple cultivars (*Malus domestica*) from the Serbian market.

Cultivar	TPC (mg GAE/kg FW)	DPPH IC50 (μg/mL)	FRAP (mg AAE/kg FW)
**Gala ***	620.67 ± 57.55 ^ab^	360.94 ± 88.65 ^a^	210.41 ± 42.19 ^a^
**Red Jonaprince**	972.74 ± 182.74 ^d^	399.35 ± 90.15 ^a^	347.29 ± 95.26 ^a^
**Red Delicious**	844.98 ± 13.26 ^cd^	302.67 ± 9.66 ^a^	267.22 ± 26.81 ^a^
**Fuji**	696.61 ± 24.74 ^abc^	435.26 ± 40.25 ^a^	311.18 ± 44.11 ^a^
**Granny Smith**	579.39 ± 24.28 ^a^	1182.17 ± 275.04 ^b^	300.82 ± 52.30 ^a^
**Idared**	807.63 ± 6.56 ^bcd^	349.43 ± 9.68 ^a^	318.10 ± 40.76 ^a^
**Golden Delicious**	692.12 ± 7.75 ^abc^	439.95 ± 50.76 ^a^	214.14 ± 29.56 ^a^

* Data are expressed as means (*n* = 3) ± standard deviations. Lowercase letters within a column indicate statistically significant difference at *p* < 0.05.

**Table 3 foods-13-02291-t003:** Content of individual organic acids of the conventional apple cultivars (*Malus domestica*) from the Serbian market.

Conc. mg/kg FW	Gala	Red Jonaprince	Red Delicious	Fuji	Granny Smith	Idared	Golden Delicious
QA *	178.78 ± 79.73 ^b^	132.01 ± 27.29 ^ab^	526.77 ± 29.59 ^c^	127.30 ± 16.94 ^ab^	151.01 ± 25.27 ^ab^	76.33 ± 16.13 ^a^	88.31 ± 20.20 ^ab^
LA	69.39 ± 27.04 ^ab^	n.d.	165.25 ± 89.55 ^b^	93.05 ± 30.25 ^ab^	101.74 ± 40.03 ^ab^	86.22 ± 55.78 ^ab^	n.d.
ShA	69.35 ± 14.06 a	74.05 ± 11.73 ^a^	208.96 ± 120.63 ^a^	312.73 ± 132.77 ^a^	411.66 ± 405.67 ^a^	114.51 ± 7.08 ^a^	68.07 ± 17.10 ^a^
GA	n.d.	37.03 ± 49.62 ^a^	n.d.	n.d.	n.d.	n.d.	181.46 ± 36.12 ^b^
SuA	126.82 ± 64.23 ^a^	146.36 ± 110.31 ^a^	193.57 ± 28.07 ^a^	127.64 ± 53.78 ^a^	119.04 ± 17.80 ^a^	169.18 ± 10.56 ^a^	161.66 ± 7.72 ^a^
MA	4023.98 ± 631.65 ^ab^	3138.72 ± 687.10 ^a^	2744.00 ± 656.17 ^a^	4771.05 ± 453.67 ^bc^	6958.48 ± 484.10 ^d^	6359.65 ± 752.74 ^d^	5860.42 ± 435.24 ^cd^
FA	34.35 ± 19.07 ^a^	32.71 ± 3.01 ^a^	62.72 ± 37.33 ^a^	304.87 ± 219.17 ^b^	71.97 ± 33.62 ^a^	48.58 ± 34.91 ^a^	51.49 ± 12.78 ^a^
OA	16.54 ± 4.08 ^a^	24.40 ± 10.08 ^a^	14.36 ± 10.40 ^a^	25.82 ± 3.98 ^a^	16.25 ± 2.01 ^a^	44.40 ± 42.71 ^a^	16.91 ± 4.60 ^a^
CA	47.65 ± 5.84 ^a^	47.12 ± 10.35 ^a^	52.46 ± 43.57 ^a^	91.55 ± 29.44 ^ab^	106.01 ± 29.60 ^ab^	94.80 ± 11.84 ^ab^	131.87 ± 10.45 ^b^
P	118.66 ± 25.08 ^ab^	102.34 ± 38.61 ^a^	136.98 ± 38.49 ^ab^	238.18 ± 56.27 ^b^	195.22 ± 106.25 ^ab^	151.25 ± 6.00 ^ab^	248.49 ± 27.61 ^b^

* Data are expressed as means (*n* = 3) ± standard deviations. Lowercase letters within a row indicate statistically significant differences at *p* < 0.05. Abbreviations: QA—quinic acid; LA—lactic acid; ShA—shikimic acid; GA—galacturonic acid; SuA—succinic acid; MA—malic acid; FA—fumaric acid; OA—oxalic acid; CA—citric acid; P—phosphate.

**Table 4 foods-13-02291-t004:** Content of individual sugars and sugar alcohols of the conventional apple cultivars (*Malus domestica*) from the Serbian market.

Conc. g/kg FW		Gala	Red Jonaprince	Red Delicious	Fuji	Granny Smith	Idared	Golden Delicious
Sugars *	Glu	20.95 ± 0.70 ^abc^	24.42 ± 4.92 ^c^	21.11 ± 1.68 ^bc^	21.89 ± 1.77 ^bc^	16.29 ± 0.87 ^ab^	18.81 ± 1.26 ^abc^	14.64 ± 1.85 ^a^
	Fru	54.43 ± 3.84 ^c^	52.87 ± 9.41 ^c^	40.52 ± 2.92 ^ab^	49.89 ± 3.37 ^bc^	34.60 ± 1.30 ^c^	55.61 ± 1.51 ^c^	52.62 ± 4.47 ^c^
	Suc	15.84 ± 0.53 ^a^	20.04 ± 4.59 ^ab^	22.20 ± 2.20 ^ab^	22.77 ± 1.52 ^ab^	24.77 ± 1.17 ^bc^	28.45 ± 4.70 ^bc^	32.92 ± 2.41 ^c^
	Gal	n.d.	n.d.	n.d.	n.d.	n.d.	n.d.	n.d.
Sugar alcohols	S	2.61 ± 0.52 ^a^	3.62 ± 1.56 ^a^	3.47 ± 0.20 ^a^	2.87 ± 0.64 ^a^	2.31 ± 1.55 ^a^	3.57 ± 0.43 ^a^	3.58 ± 0.09 ^a^
	MI	0.59 ± 0.12 ^ab^	0.48 ± 0.06 ^a^	0.82 ± 0.05 ^c^	0.47 ± 0.10 ^a^	0.81 ± 0.04 ^c^	0.69 ± 0.10 ^bc^	0.85 ± 0.11 ^c^

* Data are expressed as means (*n* = 3) ± standard deviations. Lowercase letters within a row indicate statistically significant differences at *p* ≤ 0.05. Abbreviations: Glu—glucose; Fru—fructose; Suc—sucrose; Gal—galactose; S—sorbitol; MI—myo-inositol.

**Table 5 foods-13-02291-t005:** Correlation factors between phenolics content and antioxidant activity.

Phenolics	DPPH	FRAP
Arbutin	−0.288	0.292
Gallic acid	−0.236	−0.025
Neo-chlorogenic acid	−0.08	−0.561
Catechin	0.745	−0.009
Chlorogenic acid	−0.709	−0.189
*p*-coumaric acid	−0.853 *	−0.317
Procyanidin B2	0.834 *	−0.054
Epicatechin	0.737	−0.021
Qu-3-galactoside	0.236	0.209
Qu-3-glucoside	0.716	0.363
Phloridzin	−0.656	−0.364
Qu-3-rutinoside	0.039	0.126
Qu-3-rhamnoside	−0.048	0.286
Total polyphenols content	−0.462 *	0.543 *

* Correlations are significant at *p* < 0.05.

**Table 6 foods-13-02291-t006:** The weight coefficients and biases, *W*_1_ and *B*_1_, for the ANN model.

	1	2	3	4	5	6	7	8	9	10	11
A *	0.055	0.184	−0.187	0.839	0.337	0.062	0.114	0.217	0.214	0.178	0.335
GA	0.663	−0.232	0.355	0.748	0.278	0.230	−0.118	0.494	−0.062	−0.835	−0.301
C	0.021	−0.277	0.330	−0.222	−0.196	0.169	−0.083	0.242	0.163	−0.224	−0.148
CA	0.124	−0.260	0.006	0.003	−0.214	0.101	−0.221	−0.272	−0.378	−0.116	−0.490
pCA	0.018	−0.314	−0.248	−0.136	−0.025	−0.303	−0.185	−0.502	−0.353	−0.024	0.008
pCB2	−0.488	0.027	−0.009	−0.886	−0.339	−0.144	0.114	−0.100	0.103	0.399	0.351
E	−0.210	−0.042	0.113	−0.605	−0.171	−0.016	0.100	0.072	0.027	−0.014	0.266
Q3ga	0.009	0.110	−0.063	0.520	0.060	0.162	0.127	0.369	0.301	−0.013	0.189
Q3gl	−0.174	0.233	0.230	−0.133	−0.215	0.396	0.247	0.552	0.198	0.003	−0.097
P	−0.235	0.040	−0.384	−0.537	−0.234	−0.399	−0.193	−0.770	−0.482	0.223	0.258
Q3ru	−0.215	0.261	−0.144	0.243	−0.168	0.031	0.094	0.033	0.042	0.274	0.016
Q3rh	−0.231	0.351	−0.342	0.396	0.140	−0.041	0.306	0.083	0.203	0.374	0.436
Bias	−0.080	−0.120	0.022	−0.162	−0.067	0.006	−0.010	−0.038	−0.145	−0.157	0.099

* A—arbutin; GA—gallic acid; nCA—neo-chlorogenic acid; C—catechin; CA—chlorogenic acid; pCA—*p*-coumaric acid; pCB2—procyanidin B2; E—epicatechin; Q3ga—Qu-3-galactoside; Q3gl—Qu-3-glucoside; P—phloridzin; Q3ru—Qu-3-rutinoside; Q3rh—Qu-3-rhamnoside.

**Table 7 foods-13-02291-t007:** The weight coefficients and biases, *W*_2_ and *B*_2_, for the ANN model.

	1	2	3	4	5	6	7	8	9	10	11	Bias
DPPH	−0.193	−0.669	0.025	−0.005	0.357	0.025	−0.198	0.227	0.532	0.423	−0.074	−0.118
FRAP	0.438	−0.143	0.419	0.598	−0.262	0.269	−0.126	0.371	0.010	−0.493	−0.620	0.152

## Data Availability

All data generated or analyzed during this study are available from the corresponding author upon reasonable request.

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
