# Peer review of "Assessing Antioxidant Properties, Phenolic Compound Profiles, Organic Acids, and Sugars in Conventional Apple Cultivars (Malus domestica): A Chemometric Approach"

_foods, 2024, doi:10.3390/foods13142291_

Round 1

Reviewer 1 Report

Comments and Suggestions for Authors

The manuscript “Assessing Antioxidant Properties, Phenolic Compound Profiles, Organic Acids and Sugars in Conventional Apple Cultivars (Malus domestica): a Chemometric Approach” by Cvetković et al. attempts to analyze the basic quality characteristics, antioxidant activity and phenolic composition of 7 apple varieties available on the Serbian market.

Unfortunately, the article does not contain any news. This was also not explained in terms of the purpose of the work. The introduction to the work contains very basic information about the crops and chemical composition of apples. The issues of the manuscript were not presented. Moreover, line 73-81 included in the introduction should not be included there. I am not sure whether the authors checked the article thoroughly before submitting it to the editor.

The material and methods section is correctly described, but the selected apple evaluation methods are very commonly performed. Moreover, methods of assessing antioxidant activity such as DPPH and FRAP are no longer recommended due to their very low reflection in the system at the cellular, human or animal level.

In the results and discussion chapter, the authors describe the obtained data, but do not refer them to the available literature data. Hence, and at this level, it is difficult to say where the novelty of the submitted manuscript lies. Have the selected apple varieties not already been evaluated in other countries? In the PCA analysis and correlation of the obtained results, the authors describe the data without explaining what they may result from. The negative correlation between TPC and DPPH is likely due to the fact that the results for the DPPH method are presented as IC50 values. Before correlating them with other results, they must be formatted appropriately.

Unfortunately, the submitted article requires thorough editing.

Comments on the Quality of English Language

Moderate editing of English language required. 

Author Response

The manuscript “Assessing Antioxidant Properties, Phenolic Compound Profiles, Organic Acids and Sugars in Conventional Apple Cultivars (Malus domestica): a Chemometric Approach” by Cvetković et al. attempts to analyze the basic quality characteristics, antioxidant activity and phenolic composition of 7 apple varieties available on the Serbian market.

Comment 1: Unfortunately, the article does not contain any news. This was also not explained in terms of the purpose of the work. The introduction to the work contains very basic information about the crops and chemical composition of apples. The issues of the manuscript were not presented.

Response 1: Thank you for pointing that out. The introduction part has been improved with the chemical composition of certain apple varieties. The issue of the manuscript is pointed out more clearly in the last paragraph of the Introduction part.

Comment 2: Moreover, line 73-81 included in the introduction should not be included there. I am not sure whether the authors checked the article thoroughly before submitting it to the editor.

Response 2: Thank you for your observation. This part has been deleted from the manuscript. 

Comment 3: The material and methods section is correctly described, but the selected apple evaluation methods are very commonly performed. Moreover, methods of assessing antioxidant activity such as DPPH and FRAP are no longer recommended due to their very low reflection in the system at the cellular, human or animal level.

Response 3: Thank you for your comment. The authors agree with the reviewer that DPPH and FRAP methods of assessing antioxidant activity cannot be directly comparable with antioxidant activity at the cellular, human, or animal levels. However, they were chosen because they are the most commonly applied methods and thus enable readers to easily compare the results with other studies.

In this sense, the discussion is expanded with a comparison with other studies (Lamperi et al., 2008)

Comment 4: In the results and discussion chapter, the authors describe the obtained data, but do not refer them to the available literature data. Hence, and at this level, it is difficult to say where the novelty of the submitted manuscript lies. Have the selected apple varieties not already been evaluated in other countries? In the PCA analysis and correlation of the obtained results, the authors describe the data without explaining what they may result from. The negative correlation between TPC and DPPH is likely due to the fact that the results for the DPPH method are presented as IC50 values. Before correlating them with other results, they must be formatted appropriately.

Response 4: Thank you for your comment. The selected apple varieties were analyzed in other countries, but only the content of phenolic compounds or only the content of sugars and organic acids, not together. Artificial neural network (ANN) was used only in one case (Çetin & SaÄŸlam, 2022)  to predict the antioxidant activity of apples, and it did not use individual phenolic compounds as an input and only 3 different cultivars. The authors agree with the reviewer that the negative correlation between TPC and DPPH is because the results for the DPPH method are presented as IC50 values, and this explanation has been added to the text.  The results of the DPPH test cannot be presented in another format since no standard, such as Trolox, was used in the test, and therefore results cannot be recalculated from IC50 values to equivalents per fresh weight. However, a negative correlation also can show a tendency and give rise to conclusions regarding the effect of TPC on antioxidant activity.

In the PCA analysis and correlation of the obtained results, the authors describe the data without explaining what they may result from.

As it was mentioned in the article, the PCA analysis of apple samples' phenolic content (Figure 1) revealed that the first two principal components accounted for 65.4% of the total variability among the 13 parameters (A – arbutin; GA - gallic acid; nCA - neo-chlorogenic acid; C – catechin; CA - chlorogenic acid; pCA - p-coumaric acid; pCB2 - procyanidin B2; E – epicatechin; Q3ga - Qu-3-galactoside; Q3gl - Qu-3-glucoside; P – phloridzin; Q3ru - Qu-3-rutinoside; Q3rh - Qu-3-rhamnoside).

In this analysis, certain phenolics such as procyanidin B2 (17.6% contribution to total variance), epicatechin (17.3%), and quercetin-3-glucoside (8.7%) had a positive influence on the first principal component (PC1). Conversely, chlorogenic acid (13.2%) and p-coumaric acid (14.3%) negatively influenced PC1. For the second principal component (PC2), phenolics like arbutin (19.6% of total variance), quercetin-3-galactoside (19.2%), quercetin-3-rutinoside (24.2%), and quercetin-3-rhamnoside (26.1%) showed a positive effect.

The 'Red Jonaprince' apple cultivar was distinguished by its high content of arbutin and quercetin-3-rhamnoside. 'Red Delicious' stood out with the highest gallic acid and high epicatechin content, while 'Granny Smith' had the highest levels of epicatechin, catechin, and procyanidin B2. Despite their different colors, 'Idared,' 'Golden Delicious,' and 'Fuji' were closely grouped in the same quadrant due to their similar phenolic profiles.

Comment 5: Unfortunately, the submitted article requires thorough editing.

Response 5: Thank you for your useful suggestions, making this paper a better and more complete scientific publication.

Reviewer 2 Report

Comments and Suggestions for Authors

In my opinion the article is interesting and well written, but I would ask the authors to clarify one thing and make the following changes to the text.

- The results for polyphenols (HPLC), organic acids and sugars are converted to 1 kg of fresh fruit weight, while the results for total phenolics and antioxidant activity are expressed per 1 g of freeze-dried fruit. For a statistical analysis in which the above parameters are compiled (compared) (e.g. Fig. 5, Tab. 6), should not the results obtained be expressed in the same unit (either per dry weight or per fresh weight)? The study of the relationship between variables may give contradictory results if the results have a 'mixed' unit, as the water content of the varieties tested may be different (freeze-drying does not remove water completely). Therefore, I believe that the authors should convert all results to dry weight or convert only the results from antioxidant activity and total phenolic content to fresh weight, and then subject them to an appropriate statistical analysis. 

- Please insert subscripts in the sum formulas of the compounds. 

Author Response

Comment 1: In my opinion the article is interesting and well written, but I would ask the authors to clarify one thing and make the following changes to the text.

Response 1: Thank you for your comment and observation.

Comment 2: The results for polyphenols (HPLC), organic acids and sugars are converted to 1 kg of fresh fruit weight, while the results for total phenolics and antioxidant activity are expressed per 1 g of freeze-dried fruit. For a statistical analysis in which the above parameters are compiled (compared) (e.g. Fig. 5, Tab. 6), should not the results obtained be expressed in the same unit (either per dry weight or per fresh weight)? The study of the relationship between variables may give contradictory results if the results have a 'mixed' unit, as the water content of the varieties tested may be different (freeze-drying does not remove water completely). Therefore, I believe that the authors should convert all results to dry weight or convert only the results from antioxidant activity and total phenolic content to fresh weight, and then subject them to an appropriate statistical analysis. 

Response 2: The authors would like to thank the reviewer for pointing out this. It appeared in the Material and Methods, sections  2.5 and 2.6.2.  that the results for total phenolics and antioxidant activity were expressed per 1 g of freeze-dried fruit by mistake. Actually, before statistical analysis, all the measurements were converted per kg of fresh weight (FW) and that can be seen in Table 2.

Comment 3: Please insert subscripts in the sum formulas of the compounds. 

Response 3: All subscripts has been inserted.

Reviewer 3 Report

Comments and Suggestions for Authors

Please provide the origin of plant material. In to the methodology the sources is not clearly. From the market provenience could be from Serbia or other countries apple producing countries.

Line 86 please explain why you used the liquid nitrogen freezing extraction method instead of other methods such as ultrasonic extraction, microwave and steam distillation.

Line 268 please describe the reason for determination of sugar alcohols. 

I congratulate the authors for Modeling artificial neural network to predict antioxidant activity based on input parameters. 

Author Response

Comment 1: Please provide the origin of plant material. In to the methodology the sources is not clearly. From the market provenience could be from Serbia or other countries apple producing countries.

Response 1: Thank you for pointing it out. Part Materials have been corrected with information on the apple’s origin.

Comment 2: Line 86 please explain why you used the liquid nitrogen freezing extraction method instead of other methods such as ultrasonic extraction, microwave and steam distillation.

Response 2: As part of an ongoing project to analyse the phenol content of red-fleshed apple varieties and also as part of completed projects (analysis of blueberries, for example), various extraction methods were tested, with the method described here proving to be the best in terms of recovery and reproducibility.  Information on reproducibility and recovery can be found in the method by Wendelin et al.

Comment 3: Line 268 please describe the reason for determination of sugar alcohols.

Response 3: Thank you for that question. According to previous studies (Aprea et al., 2017), in addition to sugar, sugar alcohols also have a significant impact on the sweetness and taste of apples, and consequently on consumer acceptance. Connected to this their content was determined and compared in different cultivars.

Comment 4: congratulate the authors for Modeling artificial neural network to predict antioxidant activity based on input parameters.

Response 4: Thank you for this comment, we appreciate it.

Round 2

Reviewer 1 Report

Comments and Suggestions for Authors

Unfortunately, the article cannot be accepted in its current form. The article cites other works, comparing the obtained results. However, the differences in each case were explained in one and the same sentence. This is unacceptable. Unfortunately, the novelty of the work was also not emphasized. The purpose of the work remained unchanged.

Comments on the Quality of English Language

Moderate editing of English language required

Author Response

COMMENT: Unfortunately, the article cannot be accepted in its current form. The article cites other works, comparing the obtained results.

However, the differences in each case were explained in one and the same sentence. This is unacceptable.

AUTHORS: Thank you for your comment. The repeated sentence was included in the text by mistake, but it has now been corrected.

COMMENT: Unfortunately, the novelty of the work was also not emphasized. The purpose of the work remained unchanged.

AUTHORS: Thank you very much for your comments and suggestions. It is obvious that you are an expert in this field, and we gladly accept your comments. To make it easier for you to recognize all the changes, the authors have highlighted the changes into MS Word.

Several sentences were added to the Introduction section of the manuscript, in which the influence of the phenolic content on the antioxidant activity of the biological material was examined. As it was mentioned in the manuscript, The anticipation of antioxidant activities is directly correlated with the quantification of polyphenolic compounds in a given matrix [27]. Polyphenols, characterized by multiple phenol units, exhibit potent antioxidant properties by donating hydrogen atoms or electrons to neutralize reactive oxygen species [28]. Empirical studies consistently demonstrate a positive relationship between polyphenol concentration and overall antioxidant capacity, as evaluated by assays such as DPPH, ABTS, and FRAP [29]. This correlation is particularly significant in plant-based foods, where polyphenolic content serves as a reliable biomarker for potential health-promoting effects [30]. Similar correlations have been presented in apple's cultivars antiradical capacity determinations [31, 32]. Therefore, accurately measuring polyphenol levels is crucial for predicting the antioxidant efficacy of various dietary and pharmaceutical products [33]. To the best of our knowledge, such predictions have not been used for apple varieties so far.

Reviewer 2 Report

Comments and Suggestions for Authors

-

Author Response

Thank you for your comment.